# That Label's got Style: Handling Label Style Bias for Uncertain Image Segmentation

**Kilian Zepf, Eike Petersen, Jes Frellsen, Aasa Feragen**
Technical University of Denmark
`{kmze,ewipe,jefr,afhar}@dtu.dk`

## Abstract

Segmentation uncertainty models predict a distribution over plausible segmentations for a given input, which they learn from the annotator variation in the training set. However, in practice these annotations can differ systematically in the way they are generated, for example through the use of different labeling tools. This results in datasets that contain both data variability and differing label styles. In this paper, we demonstrate that applying state-of-the-art segmentation uncertainty models on such datasets can lead to model bias caused by the different label styles. We present an updated modelling objective conditioning on labeling style for aleatoric uncertainty estimation, and modify two state-of-the-art-architectures for segmentation uncertainty accordingly. We show with extensive experiments that this method reduces label style bias, while improving segmentation performance, increasing the applicability of segmentation uncertainty models in the wild. We curate two datasets, with annotations in different label styles, which we will make publicly available along with our code upon publication.

## 1 Introduction

Image segmentation is a fundamental task in computer vision and biomedical image processing. As part of the effort to create safe and interpretable ML systems, the quantification of segmentation *uncertainty* has thus become a crucial task as well. While different sources and therefore different types of uncertainties can be distinguished (Kiureghian & Ditlevsen, 2009; Gawlikowski et al., 2021), research has mainly focused on modelling two types: aleatoric and epistemic uncertainty. While epistemic uncertainty mainly refers to model uncertainty due to missing training data, aleatoric uncertainty arises through variability inherent to the data, caused for example by different opinions of the annotators about the presence, position and boundary of an object. Since aleatoric uncertainty estimates are directly inferred from the training data it is important that the variability in the available ground-truth annotations represents the experts' disagreement. However, in practice the annotations might vary in systematic ways, caused by differing labeling tools or different labeling instructions. Especially in settings where opinions in form of annotations are sourced from experts in different institutions, datasets can be heterogeneous in the way the labels are generated.

Even in the best of cases, in which manual annotators are given detailed instructions on how to segment objects, they will still have to make choices on how to annotate in ambiguous parts of the image. Moreover, annotators are not always carefully trained, and may not have access to the same

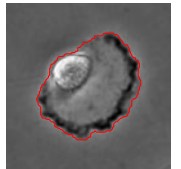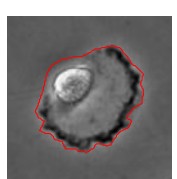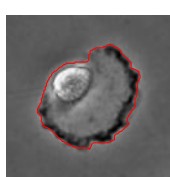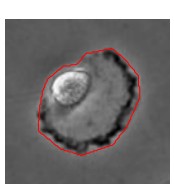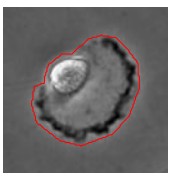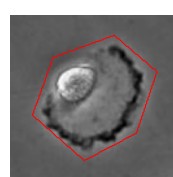

Figure 1: Sample image the from PhC-U373 dataset with annotations (red). The first three annotators were instructed to delineate the boundary in detail, whereas the last three annotators were instructed to provide a coarser and faster annotation.

labeling tools. As a result, individual choices and external factors affect how annotations are made; we term this *label style*. Figure 1 shows an example of how annotations may vary in label style.

Label style can also depend on label cost: While detailed annotations are desirable, they also take more time, and one might desire to train models on cheaper, less detailed annotations. In the example of Fig. 1, we have access to both detailed and coarse, or *weak*, annotations. It is not clear that adding the weaker annotations will necessarily improve performance; removing them to train on fewer but higher quality annotations could also be beneficial.

While weak annotations carry less precise information about the segmentation boundary, they *do* carry information about the annotator's beliefs concerning the presence and rough location of an object. Exploiting this information could improve the annotator distribution learned by the model, even tough the target might not be delineated in a detailed way. In practice, however, neither datasets nor models distinguish between variations in label style and variations in the data. As a result, current methods for segmentation uncertainty run the risk of being biased by this difference in label style.

## 1.1 CONTRIBUTION

In this paper, we demonstrate that applying state-of-the-art models on datasets that contain differing label styles can lead to systematic over-segmentation. We show how this bias can be reduced by stating an updated modelling objective for aleatoric uncertainty estimation conditioned on label style. We adjust two state-of-the-art uncertainty segmentation architectures accordingly, presenting conditioned versions of the Probabilistic U-net (Kohl et al., 2018) and the Stochastic Segmentation Networks (Monteiro et al., 2020) that fit to the updated modelling objective and can be trained on datasets containing differing label styles. We compare the proposed method against the common strategy of removing the annotations of a weaker label style from the dataset.

We curate two datasets, both with annotations in different label styles, ranging from detailed, close crops to over-segmented outlines. In a series of experiments, we show that the conditioned models outperform standard models, trained on either all or a single label style. The conditioning reduces label style bias, improves overall segmentation accuracy and enables more precise flagging of probable segmentation errors. Our results stress that including all label styles using a conditioned model enables fully leveraging all labels in a dataset, as opposed to naively excluding weaker label styles. As such, our model contributes to increasing the applicability of uncertainty segmentation models in practice. Our code and curated datasets will be made publicly available, to enable the community to further assess models for segmentation uncertainty in the scenario with differing label styles.

## 2 BACKGROUND AND RELATED WORK

Uncertainties in deep learning in general, and image segmentation in particular, can be studied under the Bayesian framework (Bishop, 2006; Kendall & Gal, 2017). Let $D = (X, A)$ be a dataset of $N$ images $x_n \in X$ with $S$ pixels each, where each image $x_n$ is associated with $k$ ground-truth annotations $a_n^k \in A$, drawn from the unknown annotator distribution $p(a|x_n)$. Furthermore, let $f(x, \theta)$ denote a model of $p(a|x)$ defined by parameters $\theta$. Formulating the segmentation task in a Bayesian way, we seek to model the probability distribution $p(y|x)$ over model predictions $y$ given an image $x$ to be as similar as possible to the annotator distribution $p(a|x)$. This predictive distribution can be decomposed into the two types of uncertainty (Kiureghian & Ditlevsen, 2009) as follows:

$$p(y|x, D) = \int \underbrace{p(y|x, \theta)}_{\text{aleatoric uncertainty}} \underbrace{p(\theta|D)}_{\text{epistemic uncertainty}} \, \mathrm{d}\theta. \qquad (1)$$

After observing the data $D$ during training, the posterior distribution $p(\theta|D)$ describes a density over the parameter space of the model, capturing epistemic uncertainty. The distribution $p(y|x, \theta)$, on the other hand, captures the variation in the data and possible model predictions, i.e., aleatoric uncertainty. Due to the typically intractable epistemic uncertainty distribution, the integral on the right hand side of equation 1 is usually not accessible. Therefore, it is of particular interest to develop suitable approximations of the predictive distribution or parts of the integral in 1, and various image segmentation approaches and models have been proposed for this purpose (Kohl et al., 2018; Monteiro et al., 2020; Kohl et al., 2019; Baumgartner et al., 2019).

In this context, the standard cross-entropy minimization approach pursued in most current deep learning research can be understood as approximating the posterior distribution $p(\theta|D)$ by a Dirac distribution $\delta(\theta - \theta_1)$ and assuming that there is no spatial correlation between the pixels in an image. Under these assumptions, one obtains

$$-\log p(a|x, D) = -\log p(a|x, \theta_1) = -\log \prod_{i=1}^{S} p(a_i|x_i, \theta_1) = -\sum_{i=1}^{S} \log p(a_i|x_i, \theta_1), \quad (2)$$

which is precisely the standard negative log likelihood (or cross-entropy) loss.

Variational Bayesian methods approximate the intractable integrals arising in Bayesian inference directly through optimization. A special case that uses Bernoulli distributions to approximate the posterior distribution of the parameters as well as the predictive distribution is the Monte Carlo Dropout method (Gal & Ghahramani, 2016a; 2015). Pixel-wise uncertainty values can be retrieved by averaging multiple forward passes, while applying dropout during inference time before each weight layer of a neural network. The resulting approximation of the predictive distribution is always multi-modal and not necessarily expressive (Folgoc et al., 2021), and drawn samples can lack coherence (Czolbe et al., 2021; Gal & Ghahramani, 2016b).

Ensemble methods (Lakshminarayanan et al., 2017) also yield pixel-wise uncertainty estimates by averaging over the predictions of different models which are independently trained on the same dataset. Contrary to Monte Carlo Dropout, which implicitly averages multiple models during inference time, those models do not share the same weights but can be seen as a frequentist approach to estimating $p(\theta|D)$. On the other hand, ensembles and multi-head models (Rupprecht et al., 2017; Lee et al., 2016) independently trained on one expert's annotation each return – in expectation – a distribution over annotations, therefore also modelling the variability and disagreement between the annotators $p(a|x)$ (Czolbe et al., 2021).

Models that mainly focus on modelling the annotator distribution $p(a|x)$ have been introduced based on combinations of deterministic segmentation networks and generative modelling techniques such as conditional variational autoencoders and normalizing flows (Baumgartner et al., 2019; Kohl et al., 2019; Selvan et al., 2020). Moreover, a Gaussian-Process based convolutional architecture was recently suggested to distinguish between annotator variability and estimator uncertainty (Popescu et al., 2021). Other approaches model the distribution over annotations with probabilistic graphical models and combinations of those with neural networks (Batra et al., 2012; Kirillov et al., 2015a;b; 2016; Arnab et al., 2018; Kamnitsas et al., 2017). However, the computational expense of inference in those models usually prohibits more than a maximum a posteriori estimate of the targeted distribution.

In the following, we will describe in detail two state-of-the-art methods for quantifying aleatoric segmentation uncertainty: the probabilistic U-net (Kohl et al., 2018) and the Stochastic Segmentation Networks (Monteiro et al., 2020). We propose a new modelling objective for the annotator distribution $p(a|x)$ including label styles, and we show how both models can be modified to fit it, increasing their applicability to datasets with varying label styles.

## 3 SEGMENTATION UNCERTAINTY MODELS CONDITIONED ON LABEL STYLE

Segmentation uncertainty models fit a predictive distribution $p(y|x, \theta)$ to the annotator distribution $p(a|x)$. We argue that, in practice, the targeted distribution of annotations should also be conditioned on label style. Therefore, we propose to model $p(a|x, l)$ instead, where $l \in \{0, \ldots, i > 0\}$ denotes a discrete variable representing the label style, and $i$ denotes the number of available label styles. In this setting, the segmentation model is to be learned from a dataset $D = (X, A, L)$ containing tuples $(x_n, a_n^k, l_n^k)$ of images $x_n \in X$, annotations $a_n^k \in A$, and corresponding label styles $l_n^k \in L$. In the following, we present modified versions of the probabilistic U-net and Stochastic Segmentation Networks that incorporate label styles directly into training and inference by conditioning the models on a discrete variable (Mirza & Osindero, 2014), therefore fitting the new modelling objective.

### 3.1 CONDITIONED PROBABILISTIC U-NET

The probabilistic U-net (Kohl et al., 2018) combines a U-net with a conditional variational auto-encoder. To encode plausible segmentation variants, an encoder $P$ parameterized by $\omega$ takes an

image $x$ as its input and estimates the mean $\mu_\omega(x)$ and variance $\sigma_\omega(x)$ of a diagonal Gaussian $\mathcal{N}(\mu_\omega(x), \sigma_\omega(x))$ in $\mathbb{R}^6$. During inference, a sample $z$ from this distribution is drawn and concatenated with the deterministic output of the U-net $g_\theta(x)$ and combined by $1 \times 1$ convolutions $f_\psi$. A prediction $y$ for a given latent $z$ can then be written as

$$y = f(g_\theta(x), z, \psi). \tag{3}$$

The prior net is trained by minimizing the Kullback-Leibler divergence between its predicted latent space distribution $P$ and a distribution given by the encoder $Q$, called the posterior net. $Q$ has parameters $\nu$ and receives both the image $x$ and annotation $a$ as inputs, estimating the parameters of the distribution $\mathcal{N}(\mu_\nu(x, a), \sigma_\nu(x, a))$. We adjust this model by adding a discrete label style variable $l$ to the input of the prior net encoder, which is tiled and concatenated one-hot-encoded to the channel axis of the input image $x$. The latent variable $z$ is then modelled as the normal distribution

$$z|x, l \sim \mathcal{N}(\mu_\omega(x, l), \sigma_\omega(x, l)), \tag{4}$$

where the covariance matrix $\sigma_\omega$ is, again, assumed to be diagonal, and both $\sigma_\omega$ and $\mu_\omega$ are estimated by the prior net encoder. During training, the posterior net $Q$ receives the style $l_n^k$ in addition to the image $x_n$ and annotation $a_n^k$. Like the original model, the architecture is trained by minimizing the variational lower bound as described in Kohl et al. (2018) (see appendix A.5 for details).

## 3.2 Conditioned Stochastic Segmentation Networks

Stochastic Segmentation Networks (SSN) (Monteiro et al., 2020) are a recently suggested model class for quantifying aleatoric segmentation uncertainty. The method can be applied to any feature map received by a deterministic segmentation network, in our case a U-net $g_\theta(x)$. The feature map is passed through three separate convolutional layers, $\mu(x)$, $D(x)$, and $P(x)$, that estimate the parameters of a low-rank multivariate normal distribution over the logits $\eta$. Since $g$ is deterministic, we can write the layers as directly dependent on $x$. The covariance matrix is given by

$$\Sigma(x) = D(x) + P(x)P(x)^T, \tag{5}$$

and one can then pass samples drawn from the estimated logit distribution

$$\eta|x \sim \mathcal{N}(\mu(x), \Sigma(x)) \tag{6}$$

through a softmax layer to receive predictions for a given image $x$. For details on the training procedure and loss function, we refer the reader to appendix A.5.

We adjust this model by passing the discrete label style variable $l$ through an encoder and by concatenating the resulting feature map to the feature map given by the U-net. The combined feature map is then passed through the respective convolutional layers that estimate the parameters of the logit distribution. The style variable is again tiled and concatenated (one-hot-encoded) to the channel axis of the feature maps. The distribution over the logits $\eta$ is then modelled as

$$\eta|x, l \sim \mathcal{N}(\mu(x, l), \Sigma(x, l)). \tag{7}$$

Given an image during test time, it is now possible with both models to condition their predictions on a label style $l \in \{0, \ldots, i > 0\}$ that has been used for training. Figure 5 in appendix A.4 gives a schematic overview of the baseline models and the conditioned versions during inference.

## 4 Experiments

To train and evaluate segmentation uncertainty models fitting the updated modelling objective $p(a|x, l)$, we need datasets with multiple annotations per image in differing label styles. We consider two such datasets: a subset of the ISIC19 dataset and a new version of the PhC-U373 dataset.

## 4.1 Data

For our first evaluation, we consider a subset of the ISIC19 skin lesion segmentation challenge (Combalia et al., 2019; Codella et al., 2018; Tschandl et al., 2018), where each image has exactly three annotations available. A distinctive feature of this dataset is the clear difference between

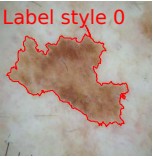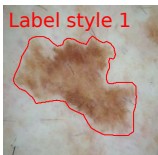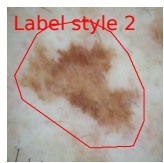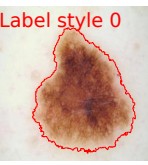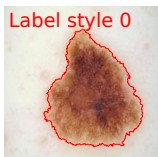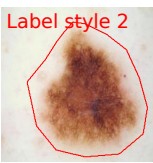

Figure 2: Two sample images from the ISIC dataset with annotations in different label styles (red): fine-grained delineations (style 0), smooth, close borders (style 1) and coarse crops (style 2). An image might have multiple annotations of a certain label style (as in the right example).

label styles, exemplified in Fig. 2: Some annotations follow the skin lesion boundary as exactly as possible; this is important as the size and boundary features can be used to clinically classify lesions as malignant or benign (Abbasi et al., 2004). Other annotations, conversely, consist of loosely defined regions containing the lesion. This type of weak labeling is more consistent with the task of detecting lesions than segmenting them.

The ISIC dataset contains ground-truth segmentations generated by three different methods, each of which we consider as one label style. Label style 0 corresponds to tight annotations with a detailed boundary, generated by a semi-automated flood-fill algorithm supervised by an expert. Label style 1 annotations stem from a fully-automated algorithm, reviewed and accepted by experts, while label style 2 are polygonal annotations traced manually by specialists. Note that not every image has exactly one annotation of each label style, so different combinations can occur such as in Fig. 2 – we include such images on purpose to illustrate our models' ability to make the most of the available data when the choice of annotations is beyond our control. We consider the fine-grained annotations of label style 0 the ground truth in the downstream analysis of annotator bias. For our experiments, the images and annotations are rescaled to $256 \times 256$ pixels.

Our second dataset is based on a cell-tracking video from the PhC-U373 dataset of the ISBI cell tracing challenge (Ulrich et al., 2009; Ulman et al., 2017). We use the first video sequence, containing 115 2D images of multiple cells, annotated with two classes (Cell, Background). To obtain images of single cells, we find the smallest bounding boxes around the ground-truth masks and extend those by 20px on all sites. We then crop these bounding boxes out of the original images and use only those patches where the expanded bounding box lies completely within the full-sized image. This results in a dataset containing 651 images of single cells, which are resized to $128 \times 128$ pixels. In addition to the ground-truth labels, all images were annotated by three researchers independently, of which two labeled the dataset twice (in different styles), resulting in five additional annotations. Labelers were instructed to perform either detailed annotation (label style 0) or wider annotations (label style 1). In total, we end up with three annotations labeled in style 0 and three annotations labeled in style 1 for each image in the dataset. Label style 0 is, again, considered the ground truth in the downstream task analysis of annotator bias later on.

Table 1: Datasets and number of image-annotation pairs contained in the different splits.

| | ISIC | | | |
| --- | --- | --- | --- | --- |
| | Train | Val | Test | Total |
| Style 0 | 62 | 20 | 22 | |
| Style 1 | 54 | 18 | 19 | |
| Style 2 | 63 | 21 | 21 | |
| All | 179 | 59 | 62 | **300** |
| | PhC-U373 | | | |
| | Train | Val | Test | Total |
| Style 0 | 1170 | 390 | 393 | |
| Style 1 | 1170 | 390 | 393 | |
| All | 2340 | 780 | 786 | **3906** |

Both datasets are divided into subsets containing only one label style and randomly split for training, validation and testing with a ratio of 60%, 20% and 20% respectively. Table 1 gives an overview over the resulting datasets and splits. Since all models are trained on pairs of images and annotations, we report this number for a fair comparison of the dataset sizes.

## 4.2 MODELS

The following models are compared in our experiments: (1) The proposed conditioned models, namely the style-conditioned probabilistic U-net (c-prob. U-net) and the style-conditioned Stochastic Segmentation Network (c-SSN), which are trained on annotations of all label styles. During inference, we condition on the style that the model will be evaluated on. (2) Probabilistic U-nets (prob. U-net) (Kohl et al., 2018) and Stochastic Segmentation Networks (SSN) (Monteiro et al., 2020) trained

on subsets that only contain one specific label style. These models are indicated by a *(subset)* tag. (3) Probabilistic U-nets and Stochastic Segmentation Networks trained on the complete dataset, containing annotations of all label styles, but not conditioned on label style. These models are indicated by an *(all)* tag. Note that this would be the most common way to use the data.

All models are implemented in PyTorch and share a U-net backbone with four encoder and decoder blocks for comparability. Each block contains three convolution layers and bilinear interpolation was used for upsampling. Dropout ($p = 0.5$) is used in the lowest-level feature map of all architectures. For the probabilistic U-nets, we chose a latent space dimension of 6 as in Kohl et al. (2018). Encoder networks in the probabilistic U-net are identical to the contraction path of the backbone U-net. For the stochastic segmentation networks, the output of the last decoder block of the backbone U-net is passed into three different $1 \times 1$ convolutional layers to estimate the low-rank approximation of the normal distribution. Refer to appendix A.1 for further details on the training procedure.

# 5 RESULTS

Both datasets considered in this paper contain annotations of high quality that closely outline the object of interest and ones of lower quality that generally over-segment the object of interest. In Figure 3 and Table 2, we show the distribution of area differences (measured in number of pixels) between the different models' predictions and the high-quality annotations of a fixed label style 0 test set. Intuitively, training the prob. U-net and the SSNs on all annotations leads to predictions that are biased towards too large annotations compared to the high-quality segmentations of label style 0 on both datasets. To evaluate whether it is possible to reduce this bias while maintaining the prediction quality as well as the ability of the those models to fit the annotator distribution, we compare the prob. U-net and SSNs in all experiments below against the two alternatives described above in section 4.2: Firstly, against both models trained on subsets and, secondly, against the conditioned versions that we proposed to fit the new objective for the annotator distribution $p(a|x, l)$.

Table 2: Average area difference with standard deviation in pixels between model predictions and respective label style 0 ground-truth annotations achieved on both datasets.

|  | Dataset | |
| --- | --- | --- |
| Model | ISIC | PhC-U373 |
| Prob. U-net (all) | 6019 (13648) | 370 (402) |
| Prob. U-net (subset 0) | 4879 (15106) | 584 (534) |
| **c-prob. U-net** | **311 (8332)** | **43 (417)** |
| SSNs (all) | 5176 (15629) | 716 (1022) |
| SSNs (subset 0) | **3142 (19438)** | 1534 (2131) |
| **c-SSNs** | 3199 (12987) | **262 (558)** |

## 5.1 VANILLA MODELS HAVE AN AREA BIAS; THIS DECREASES UNDER THE UPDATED MODELLING OBJECTIVE

Figure 3 and Table 2 show that the prob. U-net and SSNs, both trained on subsets of only label style 0 annotations, tend to over-segment the targets, despite removing the coarser label style annotations. Further, we find that the conditioned models show lower area bias compared to the baselines. For the ISIC dataset, the standard SSN model trained on the label style 0 subset shows a slightly lower area bias compared to the conditioned model. The same is true for the subset-trained prob. U-net in the PhC-U373 dataset, but in both cases, the model trained on the subset shows a higher variance.

## 5.2 INCREASED PREDICTIVE PERFORMANCE AND FIT TO THE ANNOTATOR DISTRIBUTION

We evaluated predictive performance to assess how well the different models can predict in a certain label style. To this end, we calculated the Intersection over Union (IoU) between the model's mean prediction against test sets that only contain annotations of the targeted label style, indicated by the column index in Table 3. Note that for column $i$, the prob. U-nets (subsets) or SSNs (subsets) were trained only on annotations of the targeted label style $i$. Across both datasets and all label styles, the best predictive performance in terms of IoU is achieved by the style-conditioned models, with the exception of the c-SSN conditioned on label style 1 on the ISIC dataset.

In addition, we computed the area under the receiver-operating characteristic curve (AUROC) with respect to pixel-wise model predictions compared to the ground-truth segmentation mask. Table 4

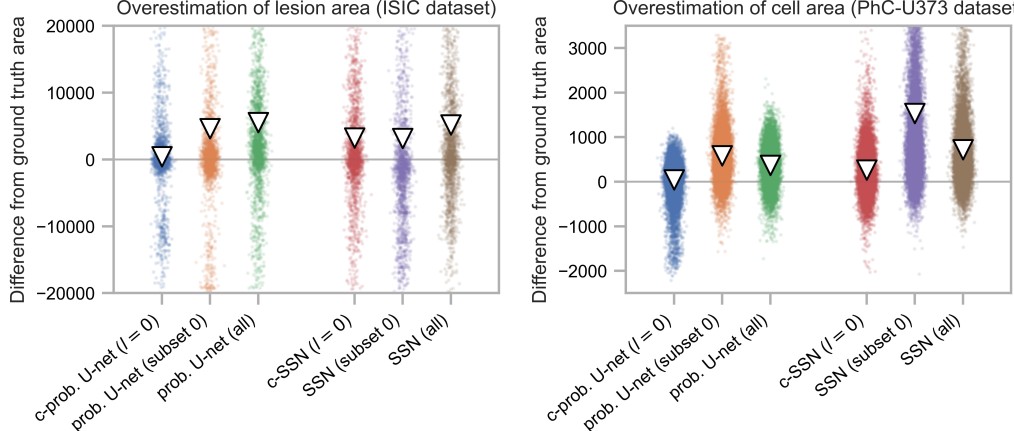

Figure 3: We assess bias in lesion and cell area estimation for the two test datasets (ISIC, PhC-U373) by distribution of area difference (in pixels) between 100 model predictions per image and respective label style 0 ground-truth annotations. Markers indicate the mean.

Table 3: Average IoU and standard deviation for the models' mean prediction on subsets of the ISIC and the PhC-U373 datasets containing only one label style each.

| | IoU wrt. to a subset of label style | | | | |
| | ISIC | | | PhC-U373 | |
| Model | 0 | 1 | 2 | 0 | 1 |
|---|---|---|---|---|---|
| Prob. U-net (all) | 0.65 (0.09) | 0.65 (0.10) | 0.69 (0.21) | 0.89 (0.01) | 0.90 (0.02) |
| Prob. U-net (subsets) | 0.66 (0.11) | 0.55 (0.17) | 0.71 (0.15) | 0.85 (0.03) | 0.86 (0.03) |
| **c-prob. U-net** | **0.77 (0.15)** | **0.78 (0.15)** | **0.76 (0.15)** | **0.92 (0.02)** | **0.92 (0.01)** |
| SSNs (all) | 0.72 (0.01) | **0.77 (0.09)** | 0.71 (0.19) | 0.89 (0.02) | **0.93 (0.02)** |
| SSNs (subsets) | 0.61 (0.13) | 0.75 (0.13) | 0.61 (0.11) | 0.85 (0.04) | 0.92 (0.02) |
| **c-SSNs** | **0.77 (0.10)** | 0.71 (0.20) | **0.78 (0.19)** | **0.92 (0.01)** | **0.93 (0.01)** |

shows the values obtained by the different models; the conditioned models show higher AUROC values across all models and label styles.

Figure 8 and Figure 9 in Appendix A.7 show qualitative results of sample predictions on 5 images from the ISIC dataset for the different models as well as the respective annotations, illustrating that the conditioning on label style corrects for the overestimation bias.

The goal of quantifying aleatoric segmentation uncertainty is formulated as fitting a model's predictive distribution $p(y|x)$ to the unknown true distribution $p(a|x)$, represented by the available annotations of ground-truth segmentations in the dataset. To assess this fit, we calculated the generalized energy distance (GED) (Székely & Rizzo, 2013; Kohl et al., 2018) between segmentations sampled from the predictive distribution of the models and the set of ground-truth segmentations. In Table 5, we show mean GED values and their standard deviation for different pairs of predictive and annotator distributions. Annotator distributions

Table 4: Pixel-wise AUROC on ISIC and PhC-U373 test sets that only contain annotations of the targeted label style, indicated by the column index.

| | ISIC | | | PhC-U373 | |
| Label style | 0 | 1 | 2 | 0 | 1 |
|---|---|---|---|---|---|
| Prob. U-net (all) | 0.9289 | 0.9102 | 0.8153 | 0.9960 | 0.9959 |
| Prob. U-net (subsets) | 0.9171 | 0.9516 | 0.8819 | 0.9925 | 0.9903 |
| **c-prob. U-net** | **0.9444** | **0.9750** | **0.9407** | **0.9963** | **0.9968** |
| SSNs (all) | 0.8873 | 0.9111 | 0.8633 | 0.9958 | 0.9950 |
| SSNs (subsets) | 0.7986 | 0.9351 | 0.8609 | 0.9893 | 0.9957 |
| **c-SSNs** | **0.9249** | **0.9506** | **0.9255** | **0.9964** | **0.9963** |

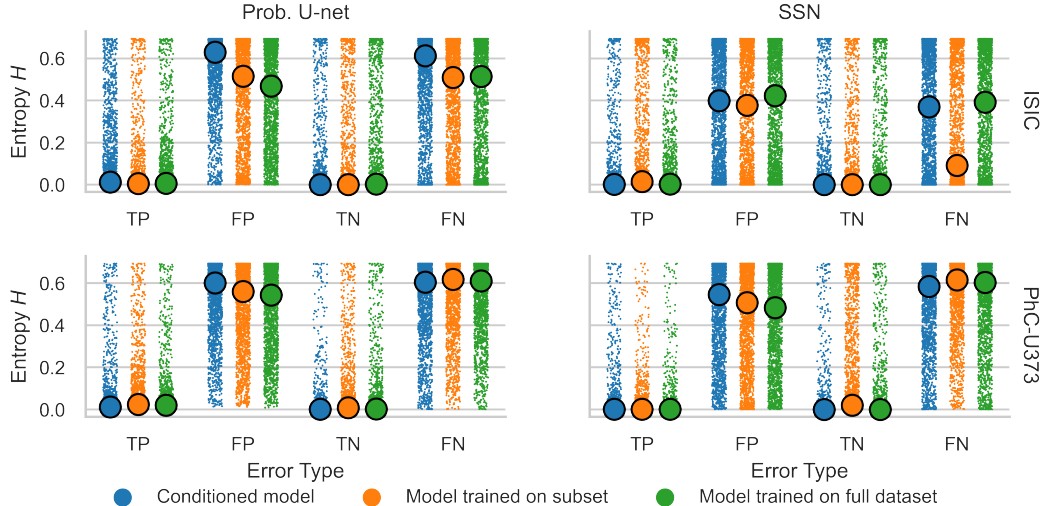

Figure 4: Distributions and medians of pixel-wise uncertainty (as quantified by the entropy of pixel-wise model predictions) for true positive (TP), false positive (FP), true negative (TN), and false negative (FN) predictions.

$p(a|x, l = i)$ contain only the set of ground-truth segmentations of label style $i$ of the test set, whereas $p(a|x)$ contains all available annotations across all label styles of the test set. Table rows contain the predictive distributions of the following models: the standard prob. U-nets and SSNs trained on all annotation styles; the standard prob. U-nets and SSNs trained on subsets of only annotation style $i$ for each column and the c-prob. U-net and c-SSN conditioned on label style $i$. We find that for the fine-grained predictions of style 0 on the ISIC dataset, the c-prob. U-net and the c-SSN exhibit lower GED values, indicating a better model fit. This also holds for the other label styles on the ISIC dataset, with the exception of the c-SSN conditioned on label style 1. On the PhC-U373 dataset, we find that the GED values are very similar across all models.

Table 5: Mean GED (with standard deviation) between models' predictive distributions and the targeted annotator distributions $p(a|x, l)$ of the available label styles as well as the full annotator distribution $p(a|x)$ of an ISIC and PhC-U373 test set.

| Annotator distribution | ISIC | | | | PhC-U373 | | |
|---|---|---|---|---|---|---|---|
| | $p(a|x, l=0)$ | $p(a|x, l=1)$ | $p(a|x, l=2)$ | $p(a|x)$ | $p(a|x, l=0)$ | $p(a|x, l=1)$ | $p(a|x)$ |
| Prob. U-net (all) | 0.58 (0.14) | 0.57 (0.23) | 0.54 (0.14) | 0.57 (0.17) | 0.69 (0.06) | **0.60 (0.07)** | **0.61 (0.06)** |
| Prob. U-net (subsets) | 0.61 (0.14) | 0.57 (0.23) | 0.51 (0.17) | - | 0.70 (0.06) | 0.62 (0.07) | - |
| **c-prob. U-net** | **0.57 (0.13)** | **0.55 (0.24)** | **0.49 (0.19)** | **0.55 (0.18)** | **0.68 (0.06)** | 0.61 (0.06) | **0.61 (0.06)** |
| SSNs (all) | 0.59 (0.14) | **0.55 (0.23)** | 0.51 (0.17) | **0.55 (0.17)** | 0.69 (0.06) | **0.60 (0.07)** | 0.61 (0.06) |
| SSNs (subsets) | 0.61 (0.15) | **0.55 (0.25)** | 0.55 (0.23) | - | 0.70 (0.06) | 0.62 (0.07) | - |
| **c-SSNs** | **0.58 (0.13)** | 0.56 (0.28) | **0.48 (0.19)** | **0.55 (0.20)** | **0.68 (0.06)** | **0.60 (0.06)** | **0.60 (0.06)** |

## 5.3 DOES HIGH SEGMENTATION UNCERTAINTY INDICATE PROBABLE SEGMENTATION ERROR?

To answer this question, we assessed the relationship between pixel-wise segmentation uncertainty and the likelihood of a segmentation error in that pixel (as compared to the ground-truth segmentation mask) on the label style 0 test set. The distribution of the pixel-wise entropy (see appendix A.3) of the predictions (as a measure of segmentation uncertainty) was assessed for the cases of true positive (TP), false positive (FP), true negative (TN), and false negative (FN) pixel predictions. A default threshold of $0.5$ was applied. The results are shown in Figure 4 for all models and both datasets. We find that the c-prob. U-net consistently assigns higher uncertainty to false positive and false negative predictions on both datasets compared to the standard prob. U-net variants. The c-SSN performs on par with the alternative models.

## 6 DISCUSSION AND CONCLUSION

Our paper took as starting point the hypothesis that probabilistic segmentation models could be biased by different label styles if these are not accounted for by the model. We find this supported by the results of Fig. 3, where we see that the prob. U-net and SSNs trained in the standard way on all label styles tend to over-segment the objects of interest. In contrast, the c-prob. U-net conditioned on the close-cropped label style 0 can reduce this bias considerably on both datasets. Even compared to prob. U-nets and SSNs trained only on the label style 0 annotations, the conditioned models give lower or on par bias.

However, the bias reduction in practice is only of value if the prediction performance does not decrease. Indeed, we find that the c-prob. U-net and the c-SSNs outperform the compared models across label styles in terms of IoU, except for the c-SSN on the label style 1 test set. On average, the mean prediction of the conditioned models is more similar to the targeted label style compared to the standard models trained on all data or on respective subsets. This indicates that the proposed conditioning of the models allows the models to implicitly correct for confounding label styles while preserve the ability to segment in a meaningful way. The strong predictive performance is supported by consistently higher AUROC values for the conditioned models in Table 4. Additionally, the conditioned models fit the annotator distributions better, as the results in Table 5 suggest. We observe that the advantage in predictive performance of the conditioned models measured by IoU is larger compared to the advantage in fitting the annotator distribution measured by GED. This might be due to the fact that the IoU is calculated on the models' mean predictions while the GED is based on 100 samples drawn from the predictive distribution.

For any segmentation uncertainty model, it is of interest to which extent the uncertainty estimates can be used to flag a high-probability segmentation error. In the context of label styles, a second highly relevant question is whether we really need fine-grained annotations, or whether we can exploit coarse-grained annotations to obtain a more precise indication of potential segmentation errors. From our preliminary analysis of the relationship between uncertainty estimates and segmentation errors, it can be observed that, as desirable, entropy is higher in the case of segmentation errors across all models and datasets (Fig. 4). The c-prob. U-net gives a consistent advantage across both datasets, while the c-SSN either improves on or performs on par with the alternative models across datasets.

To summarize, our results support that the proposed method of conditioning on label style provides an advantage over, firstly, the standard way of training a segmentation uncertainty model on all available data, ignoring label style, and, secondly, the strategy of removing confounding labels from the training set. We further demonstrate in appendix A.6 that dynamically augmenting coarse annotations from fine-grained ones does not outperform the strategy of including annotations of all label styles in a conditioned model. While we find that the standard models are biased by coarser annotations (styles 1 or 2) and the models trained on subsets might suffer from the missing training data, our proposed method enables the segmentation uncertainty model to incorporate all annotations regardless of label style in a meaningful way. This leads to the conditioned models performing best in predicting fine-grained label style 0 annotations. The modifications made to the architectures for conditioning based on the new modelling objective can be easily implemented and do not incur heavier models, while yielding better results. Finally, the ability to incorporate annotations of many different label styles into training allows for using real-world datasets as they are, thus increasing the applicability of segmentation uncertainty models to datasets as they occur in the wild.

### 6.1 LIMITATIONS AND FURTHER RESEARCH

Due to limited availability of labeled data, the correctness of the segmentation mask distributions learned by the different models can only be assessed cursorily (as done by means of the GED in Table 5). To perform a more comprehensive validation of the learned distribution, experiments with synthetic data could be performed, similar to the synthetic experiments reported by Kohl et al. (2018). More generally, an evaluation of the impact of different label styles, and the performance of the style-conditioned models, in more and larger datasets is of interest. While the conditioned models in this work require a discrete label style variable, an interesting direction for further research could be the conditioning on continuous label styles as well as considering settings in which different label styles are present but not labelled.

## ACKNOWLEDGEMENTS

This research was supported by the Novo Nordisk Foundation through the Center for Basic Machine Learning Research in Life Science (NNF20OC0062606) and the Pioneer Centre for AI, DNRF grant number P1 and Denmarks Frie Forskningsfond (9131-00097B). We want to thank Prof. Sanjay Kumar at the Department of Bioengineering, University of California at Berkeley, Berkeley CA (USA) for his permission to use his dataset 'Glioblastoma-astrocytoma U373 cells on a polyacrylamide substrate' in the context of this work. We further thank the organizers of the Cell Tracking Challenge.

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

# A  APPENDIX

## A.1  TRAINING PROCEDURE FOR THE ISIC AND PHC-U373 DATASETS

All models are trained using the Adam optimizer. For the probabilistic U-net, as well as the conditioned probabilistic U-net, we minimize the reconstruction term given by the binary cross-entropy loss, added with a weighted Kullback-Leibler divergence between the estimated normal distributions, as described by Kohl et al. (2018). The Stochastic Segmentation Networks are trained on the loss function described in Monteiro et al. (2020). Across all datasets we used a learning rate of $10^{-4}$ to train the models. For the skin lesion datasets, we trained for 600 epochs with batch size 16; for the PhC-U373 dataset, we trained for 200 epochs with a batch size of 32. The above hyperparameters were retrieved by grid search on the validation set. All computations were performed on an internal GPU cluster.

## A.2  GENERALIZED ENERGY DISTANCE

In section 5.2, we use the Generalized Energy Distance (Székely & Rizzo, 2013; Kohl et al., 2018) as a distance measure between distributions. It is defined as

$$D_{\text{GED}}^2(p, \hat{p}) = 2\mathbb{E}_{y \sim p, \hat{y} \sim \hat{p}}[d(y, \hat{y})] - \mathbb{E}_{y, y' \sim p}[d(y, y')] - \mathbb{E}_{\hat{y}, \hat{y}' \sim \hat{p}}[d(\hat{y}, \hat{y}')],$$

where $d$ is set to $1 - IoU(\cdot, \cdot)$. We set $d = 0$ if both segmentations are empty. Low GED values indicate high similarity between distributions. The expectations are approximated with 100 sample predictions from each model as done by Monteiro et al. (2020).

Note that when calculating the GED between the full annotator distribution $p(a|x)$ and the conditioned models' predictive distribution, we need to draw samples from $p(y|x, l)$. To this end, we sample $l$ from a categorical distribution with density $p(l = k) = p_k$, where $p_k$ could be set to $\frac{1}{i}$ for $i$ different label styles or be estimated form the training dataset. We choose to set $p_k = \frac{1}{3}$ for the ISIC dataset and $p_k = \frac{1}{3}$ for the PhC-U373 dataset and predict a sample prediction with the respective model, conditioned on the drawn label style.

## A.3  PIXEL-WISE ENTROPY

In section 5.3, we use the *pixel-wise* entropy as a measure of segmentation uncertainty. It is given by

$$H(p(y_{m)}|x, l)) = -p(y_m|x, l) \log p(y_m|x, l) - (1 - p(y_m|x, l)) \log(1 - p(y_m|x, l)) \tag{8}$$

for the conditioned models and

$$H(p(y_m|x)) = -p(y_m|x) \log p(y_m|x) - (1 - p(y_m|x)) \log(1 - p(y_m|x)) \tag{9}$$

for the baseline models, respectively, where $p(y_m|x, l)$ and $p(y_m|x)$ are the probabilities of being of the target class at pixel $m$ of the models' predictive distributions.

## A.4  SCHEMATIC MODEL ARCHITECTURES

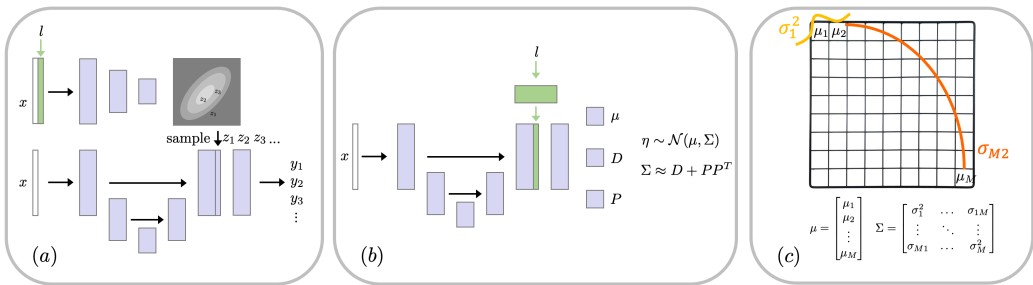

Figure 5: Schematic model architectures (adapted from Kohl et al. (2018) and Monteiro et al. (2020)) of the prob. U-net (a) and SSN (b) during inference time. Our modifications to the models are shown in green. (b) illustrates a normal distribution over the logit pixel space as predicted by a SSN.

## A.5 Loss Functions

### A.5.1 Conditioned Probabilistic U-net

For training the prob. U-net and the conditioned prob. U-net, we implemented the loss function as described by Kohl et al. (2018). While in the following we formulate the loss function for the conditioned prob. U-net case, we want to remark that the implementations of the loss functions for the conditioned and the standard model cases doe not differ. For a formulation of the loss function of the standard prob. U-net we refer the reader to Kohl et al. (2018). For a training sample $(x_n, a_n^k, l_n^k)$, prior net distribution $P$ and posterior net distribution $Q$, the loss function is calculated as

$$\mathcal{L}(a_n^k, x_n, l_n^k) = \mathbb{E}_{z \sim Q(z|x_n, a_n^k, l_n^k)} - \log p(a_n^k | y(x_n, z)) + \beta \cdot \mathrm{KL}(Q(z|x_n, a_n^k, l_n^k) || P(z|x_n))$$

$$= \underbrace{- \sum_{m=1}^{M} a_{nm}^k \log y_m(x_n, z) + (1 - a_{nm}^k) \log(1 - y_m(x_n, z))}_{\text{Binary Cross-Entropy Loss}}$$

$$+ \beta \cdot \mathrm{KL}(Q(z|x_n, a_n^k, l_n^k) || P(z|x_n))$$

for one sample from the posterior net distribution $Q$,

$$z \sim Q(z|x_n, a_n^k, l_n^k).$$

$M$ denotes the number of pixels for the images $x$ and annotations $a$. Since both $P$ and $Q$ are Gaussian, the Kullback-Leibler divergence has a closed-form solution.

### A.5.2 Conditioned Stochastic Segmentation Networks

SSNs relax the spatial independence assumption made for deriving the standard cross-entropy loss (compare equation 2) and learn a low-rank normal distribution over logits. The resulting loss function as derived in Monteiro et al. (2020) is given by

$$\mathcal{L}(a_n^k, x_n, l_n^k) = -\mathrm{logsumexp}_{s=1}^{S} \left( \sum_{m=1}^{M} \log p(a_n^k | \eta_m^{(s)}) \right) + \log(S)$$

$$= -\mathrm{logsumexp}_{s=1}^{S} \left( \underbrace{\sum_{m=1}^{M} a_{nm}^k \log(\sigma(\eta_m^{(s)})) + (1 - a_{nm}^k) \log(1 - \sigma(\eta_m^{(s)}))}_{\text{Binary Cross-Entropy Loss}} \right) + \log(S),$$

where $\sigma$ denotes the softmax function. It is calculated by drawing $S$ Monte-Carlo samples from the logit distribution

$$\eta | x, l \sim \mathcal{N}(\mu(x_n, l_n^k), \Sigma(x_n, l_n^k)). \tag{10}$$

The loss is backpropagated using the reparameterization trick. Again, we formulate the loss function for the conditioned case while remarking that it does not differ in implementation from the standard SSN case discussed in Monteiro et al. (2020).

## A.6 Dynamic Augmentation for fine-grained annotations

In this section, we test whether similar performance gains can be obtained by a dynamic augmentation strategy that only uses fine-grained annotations. For this experiment, we start with the ISIC training dataset that was used for the conditioned models and the baseline models trained on all label styles (not the subsets). For each image in the training set, coarse annotations are substituted by augmented versions of the fine-grained annotations of that image. If there are more than two fine-grained annotations available, one is selected randomly. The augmentation is done by applying a dilation operation followed by a Gaussian filter that smooths out the boundary. We train the c-prob. U-net and the c-SSN on this dynamically augmented training set utilizing the style labels. For comparison, we trained the baselines (prob. U-net and SSN) on this training set not using the style labels as described in section 4.2. The four new models are indicated by an *(aug)* tag. The models are then evaluated in terms of IoU, GED and area bias following the exact same experimental setup as in section 5.2.

Table 6: Average IoU and standard deviation for the models' mean prediction on subsets of the ISIC dataset containing only one label style each.

| | IoU wrt. to a subset of label style | | |
| --- | --- | --- | --- |
| Model | 0 | 1 | 2 |
| Prob. U-net (all) | 0.65 (0.09) | 0.65 (0.10) | 0.69 (0.21) |
| Prob. U-net (subsets) | 0.66 (0.11) | 0.55 (0.17) | 0.71 (0.15) |
| Prob. U-net (aug) | 0.68 (0.16) | 0.61 (0.14) | 0.53 (0.23) |
| c-prob. U-net (aug) | 0.73 (0.17) | 0.70 (0.09) | 0.69 (0.14) |
| **c-prob. U-net** | **0.77 (0.15)** | **0.78 (0.15)** | **0.76 (0.15)** |
| SSNs (all) | 0.72 (0.01) | **0.77 (0.09)** | 0.71 (0.19) |
| SSNs (subsets) | 0.61 (0.13) | 0.75 (0.13) | 0.61 (0.11) |
| SSNs (aug) | 0.69 (0.09) | 0.62 (0.12) | 0.59 (0.15) |
| c-SSNs (aug) | 0.75 (0.11) | 0.66 (0.14) | 0.68 (0.16) |
| **c-SSNs** | **0.77 (0.10)** | 0.71 (0.20) | **0.78 (0.19)** |

Table 7: Mean GED (with standard deviation) between models' predictive distributions and the targeted annotator distributions $p(a|x,l)$ of the available label styles as well as the full annotator distribution $p(a|x)$ of an ISIC test set.

| | ISIC | | | |
| --- | --- | --- | --- | --- |
| Annotator distribution | $p(a|x, l=0)$ | $p(a|x, l=1)$ | $p(a|x, l=2)$ | $p(a|x)$ |
| Prob. U-net (all) | 0.58 (0.14) | 0.57 (0.23) | 0.54 (0.14) | 0.57 (0.17) |
| Prob. U-net (subsets) | 0.61 (0.14) | 0.57 (0.23) | 0.51 (0.17) | - |
| Prob. U-net (aug) | 0.59 (0.14) | 0.61 (0.20) | 0.62 (0.13) | 0.61 (0.15) |
| c-prob. U-net (aug) | 0.59 (0.12) | 0.57 (0.21) | 0.53 (0.21) | 0.58 ( 0.10) |
| **c-prob. U-net** | **0.57 (0.13)** | **0.55 (0.24)** | **0.49 (0.19)** | **0.55 (0.18)** |
| SSNs (all) | 0.59 (0.14) | **0.55 (0.23)** | 0.51 (0.17) | **0.55 (0.17)** |
| SSNs (subsets) | 0.61 (0.15) | **0.55 (0.25)** | 0.55 (0.23) | - |
| SSNs (aug) | 0.60 (0.14) | 0.58 (0.22) | 0.55 (0.16) | 0.57 (0.11) |
| c-SSNs (aug) | 0.58 (0.14) | 0.57 (0.23) | 0.53 (0.20) | 0.57 (0.15) |
| **c-SSNs** | **0.58 (0.13)** | 0.56 (0.28) | **0.48 (0.19)** | **0.55 (0.20)** |

As seen in Figure 6, the conditioned models are still able to correct for area bias compared to the baseline models in this setting. However, segmentation and uncertainty quantification performance are reduced. The IoU decreases relatively (see Table 6) while GED increases (see Table 7). For the baseline models, area bias and segmentation performance is worse compared to all other models.

These results support our reasoning that coarse annotations contain additional information about annotator variability, while at the same time adding area bias. Figure 7 shows examples from the ISIC dataset that illustrate exactly this: The coarse annotations do not always contain the fine-grained annotations. It is, therefore, not possible to capture all the annotator variability when using only augmentations of fine-grained annotations.

## A.7 QUALITATIVE RESULTS ON THE ISIC DATASET

Figure 8 and 9 show sample images from the ISIC dataset with overlayed annotations. Fig. 8 shows predictions (threshold 0.5) of the probabilistic U-net trained on all data, the prob. U-nets trained on the label style subsets and the c-prob. U-net conditioned on the different label styles. Fig. 9 shows predictions (threshold 0.5) of the SSN trained on all data, the SSNs trained on the label style subsets and the c-SSN conditioned on the different label styles. In both figures, the last column shows an

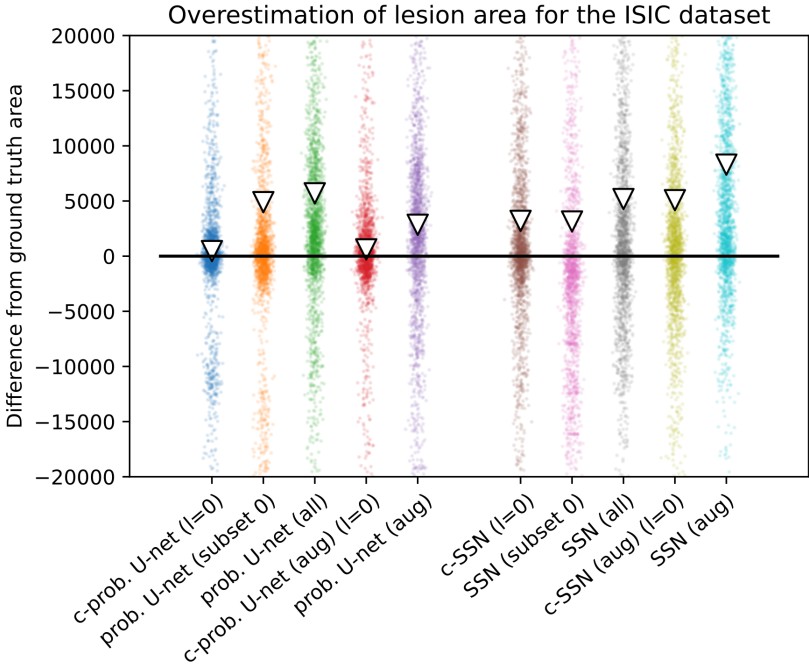

Figure 6: We assess bias in lesion and cell area estimation for the ISIC test dataset by distribution of area difference (in pixels) between 100 model predictions per image and respective label style 0 ground-truth annotations. Markers indicate the mean.

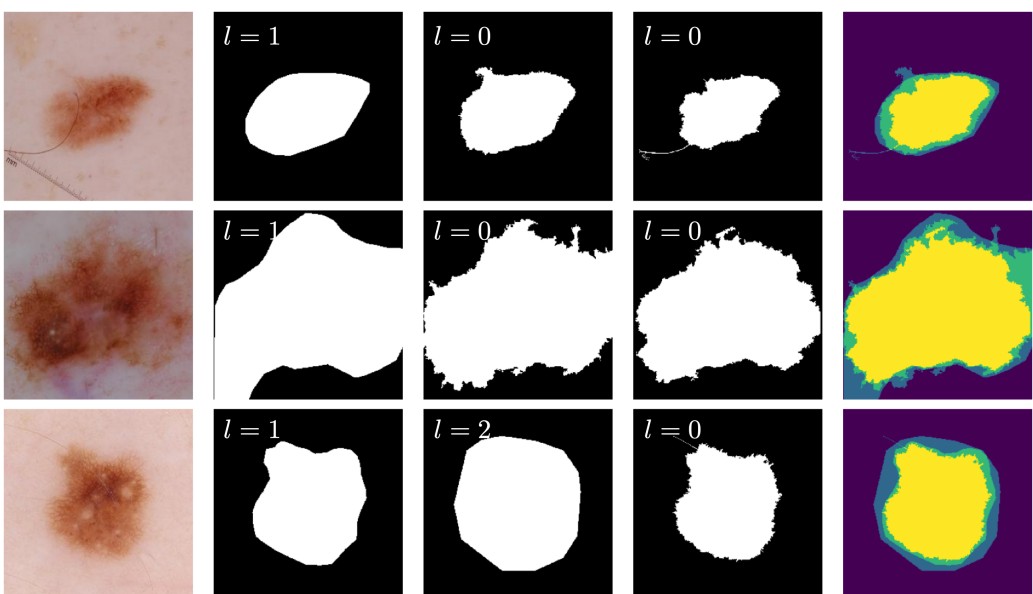

Figure 7: Three examples from the ISIC dataset with their respective annotations. The last column shows a heatmap of the average of all three annotation masks. The respective label style $l$ is written on each annotation.

overlay of the conditioned models' prediction when conditioning on each of the three label styles for the given image. In both figures, it is visible that the conditioning on label style corrects for the overestimation bias.

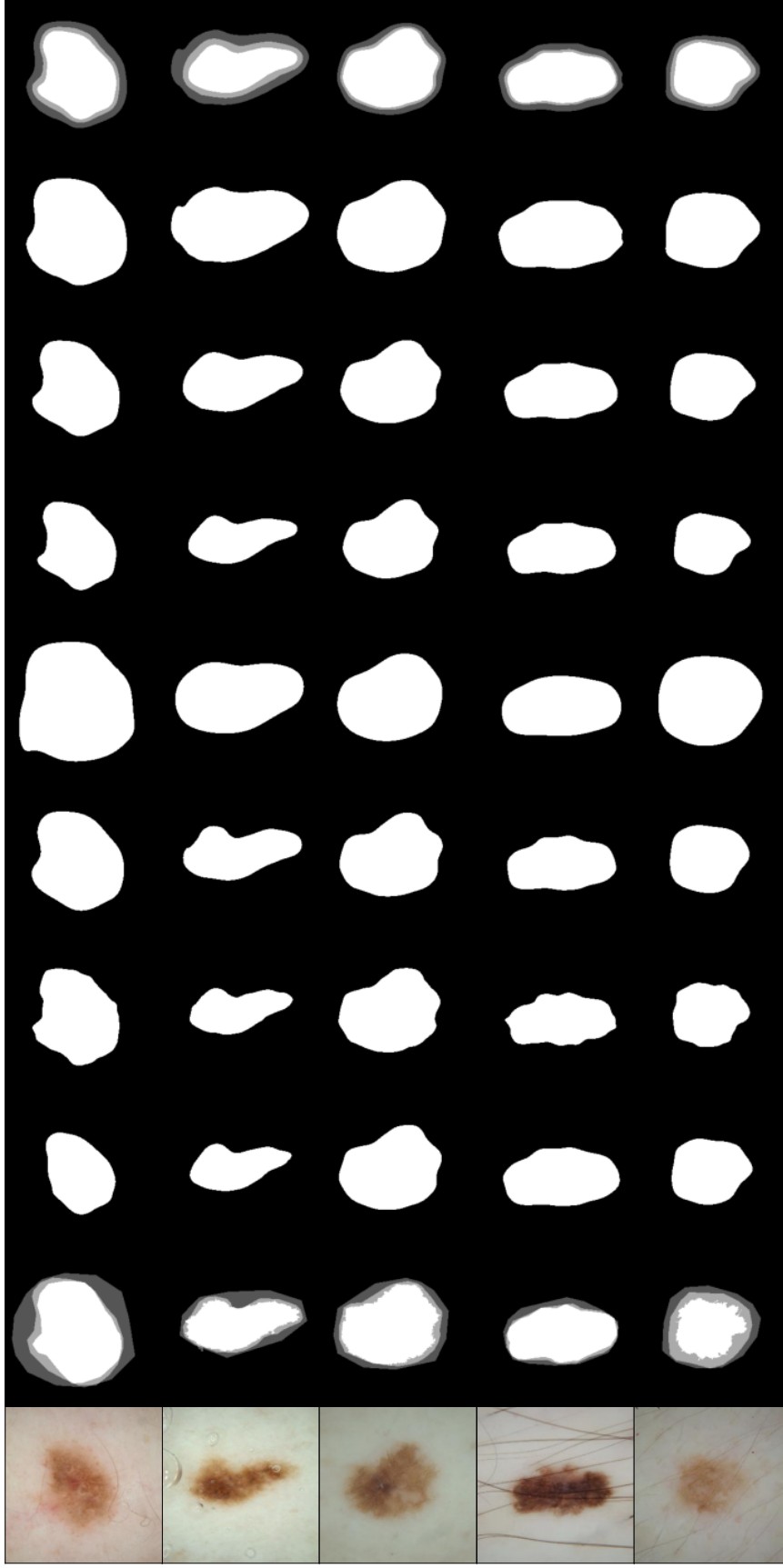

Figure 8: Mean predictions of the different prob. U-net models for 5 images from the ISIC dataset. From left to right: Image; Overlayed annotations; prob. U-net (all); prob. U-net (subset 0); prob. U-net (subset 1); prob. U-net (subset 2); c-prob. U-net conditioned on style 0; c-prob. U-net conditioned on style 1; c-prob. U-net conditioned on style 2; Overlayed predictions from c-prob. U-net for each label style.

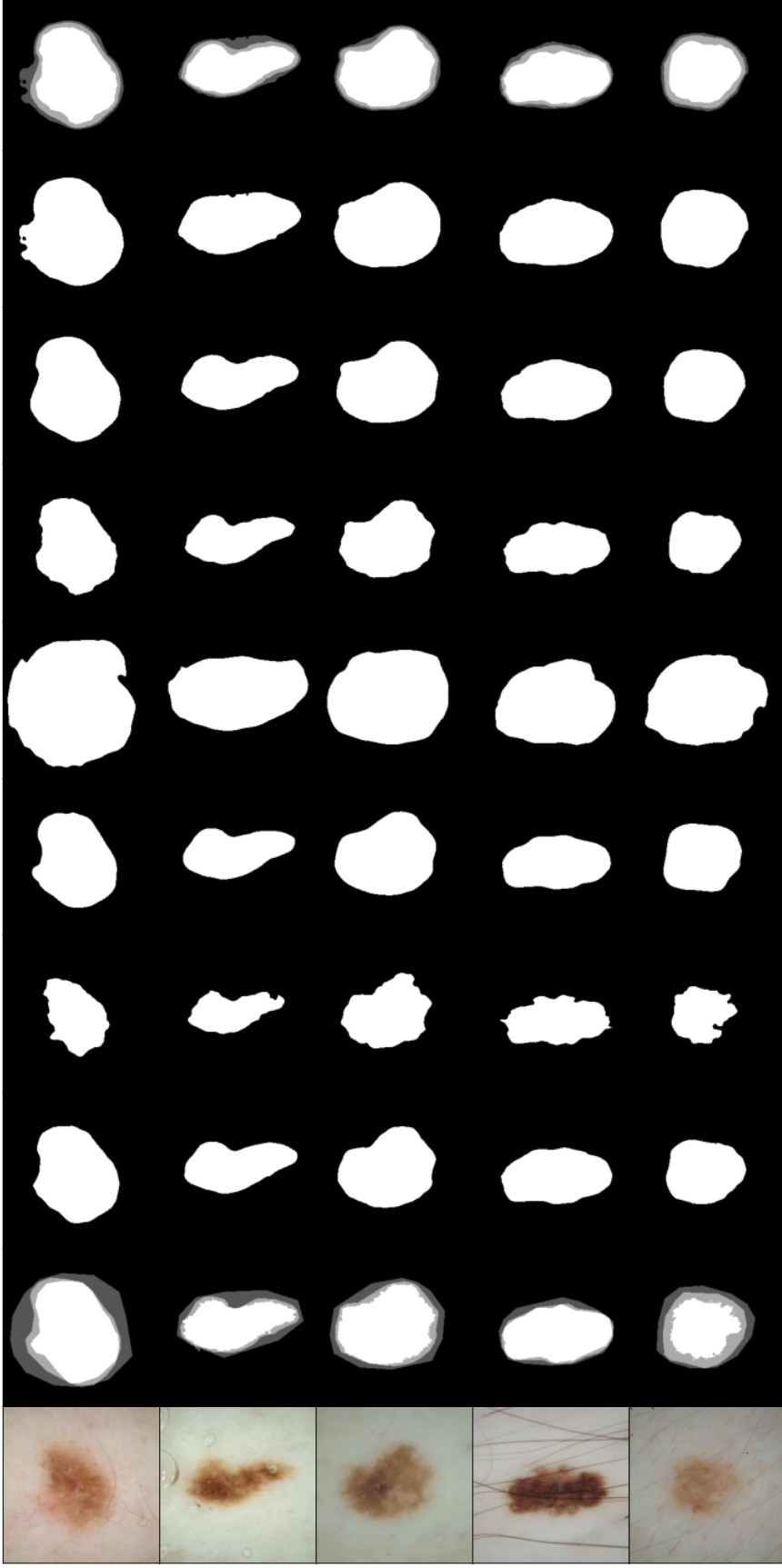

Figure 9: Mean predictions of the different SSN models for 5 images from the ISIC dataset. From left to right: Image; Overlayed annotations; SSN (all); SSN (subset 0); SSN (subset 1); SSN (subset 2); c-SSN conditioned on style 0; c-SSN conditioned on style 1; c-SSN conditioned on style 2; Overlayed predictions from c-SSN for each label style.

