# OpenReview forum: "That Label's got Style: Handling Label Style Bias for Uncertain Image Segmentation"
_ICLR.cc/2023/Conference — ICLR 2023 poster_

### Official Review · Reviewer_f94i · 2022-10-23

**Confidence:** 4
**Correctness:** 3
**Technical Novelty And Significance:** 2
**Empirical Novelty And Significance:** 3
**Recommendation:** 6

**Clarity, Quality, Novelty And Reproducibility:**

The background and theory is generally clearly written and easy to follow. The primary proposal of conditioning on style labels may not be entirely novel, but remains well-motivated and executed. The implementation of the proposed conditioning involves adding the label style variable to defined inputs of previously-described models, which should be replicable. However, some clarifications might remain on the exact training and hyperparameter optimization procedure (if any).


**Strength And Weaknesses:**

Strengths:

1. The proposed label style conditioning method allows information on annotation style to be exploited, in segmentation tasks. Moreover, as coarse segmentations are (possibly greatly) less time-consuming for human graders to produce, the method may greatly reduce human effort, for large performance gains.

2. Two model architectures are implemented, and experiments are exhaustively performed over combinations of training and testing data subsets, each containing one or all label styles.

3. The distribution of overestimations/uncertainties are well-illustrated in suitable figures.

Possible Weaknesses/Considerations:

4. A major practical consideration about the proposed method, is the requirement for the label style to be known. For the experiments studied in this work, the datasets have annotations in prior-defined styles (e.g. Figure 2), which are then separately evaluated on as subsets each containing annotations of a single style. It might firstly be informative to evaluate performance where the annotation style is not a priori known.

5. Related to the above, coarse annotations (e.g. Label Styles 1 & 2 in Figure 2) appear to be automatically obtainable from detailed annotations (Label Style 0), using some basic image processing operations (e.g. dilation, then smoothing of the boundary). If so, this may suggest that the coarse annotations actually do not really provide any (significant) new information, and it is possible that with just the detailed annotations for each image, coarse annotations can be generated and used as the additional coarse annotations for the proposed conditional learning. This would however then fall into the (common) use of dynamic augmentation (e.g. minor rotation/scaling, elastic deformation etc.) preprocessing of the image & annotation labels, in training deep learning classification/segmentation models.
Given this, it might be appropriate to explore whether automatically-generated coarse segmentations are sufficient to replicate the performance gain currently observed with manually-annotated coarse segmentations. If so, this would further inform on the actual source of the performance gain, and further reduce human annotation burden.

6. Still related to the above, while area bias (and possibly other bias) is observed (Section 5.1), it is not clear the extent to which such bias has been attempted to be mitigated, particularly using the validation data. It is mentioned in Section 4.1 that the datasets have 20% of image-annotation pairs reserved as validation data, but it is not clear how this validation data has been used. It might be expected for area (and other) bias to be mitigated, by optimizing hyperparameters such as the pixel threshold (apparently a default threshold of 0.5 is mentioned in Section 5.3) used to separate the object from the background. This might be clarified.

7. The “pixel-wise entropy” referred to in Section 5.3 might be explicitly defined (possibly in the appendix), for completeness.

8. Theoretically, the formulation p(a|x,l) appears to still place emphasis on annotator distribution, while it might be supposed that inter-annotator label differences would be insignificant compared to inter-style label differences. This might be discussed.


**Summary Of The Paper:**

This paper explores conditioning on image annotation labelling style, to reduce aleatoric uncertainty relating to the segmentation of objects in medical images. In particular, the general problem of uncertainty estimation for segmentation assumes that the annotation is drawn from an unknown annotator distribution, p(a|x) where a represents the (unknown) annotator, and x is the image. The main contribution then is the proposal to also condition on (discrete) label style l, i.e. p(a|x,l), where the label style is known. This label conditioning is implemented into two segmentation models, the (conditioned) probabilistic U-Net (CPUN), and the conditioned stochastic segmentation network (CSSN), such that both models can condition their predictions on any label style used in training. Experiments on two datasets (ISIC, PhC-U373) suggest that such conditioning results in improved segmentation performance (as measured by IoU), whether compared to training using all available labels (regardless of label style, without conditioning), or using only annotations of the same label style (thus losing information possibly obtainable from the other label styles).


**Summary Of The Review:**

A label style conditioning is proposed and implemented in two previously-described deep neural network-based image segmentation models, which exhibits improved performance on two datasets compared to the same models without conditioning. However, some issues may remain with whether the comparisons are fully fair and whether apparent bias has been attempted to be mitigated with more conventional and common techniques, as detailed in the Strength and Weaknesses section.

---

> ### Author Response · Authors · 2022-11-11
> **Response to Reviewer f94i - Part 3: Validation Data Usage**
>
> - *6. Still related to the above, while area bias (and possibly other bias) is observed (Section 5.1), it is not clear the extent to which such bias has been attempted to be mitigated, particularly using the validation data. It is mentioned in Section 4.1 that the datasets have 20% of image-annotation pairs reserved as validation data, but it is not clear how this validation data has been used. It might be expected for area (and other) bias to be mitigated, by optimizing hyperparameters such as the pixel threshold (apparently a default threshold of 0.5 is mentioned in Section 5.3) used to separate the object from the background. This might be clarified.*
>
> The validation data have been used only for performing grid search on hyperparameters, i.e., epochs, batch size, learning rate, and weight decay on the baseline models. We use the same hyperparameters across all models to ensure comparability. To clarify this, we added details to the respective appendix section “A.1 Training procedure for the ISIC and PhC-U373 datasets,” where we describe the details of the training procedure.
>
> - *7. The “pixel-wise entropy” referred to in Section 5.3 might be explicitly defined (possibly in the appendix), for completeness.*
>
> As suggested, we added a section “Pixel-wise entropy” to the appendix (A.3), stating the formulas used and providing more details on the calculation.
>
> - *8. Theoretically, the formulation p(a|x,l) appears to still place emphasis on annotator distribution, while it might be supposed that inter-annotator label differences would be insignificant compared to inter-style label differences. This might be discussed.*
>
> Indeed, the focus in our formulation $p(a|x,l)$ is still on the annotator distribution, and we would like to clarify this. The primary goal of training an aleatoric segmentation uncertainty model is to fit the true annotator distribution. We argue that the annotations can vary both in different opinions of annotators (resulting in differing masks) and in the label styles in which the annotations are delivered. The result is that variability between the annotation masks can not easily be mapped to either being caused by the differing label styles or by the disagreement of annotators – it must be learned by the model.
>
> Thank you for your valuable comments! We are happy to discuss this further and hope to have addressed your comments satisfactorily.

---

> ### Author Response · Authors · 2022-11-11
> **Response to Reviewer f94i - Part 2: Dynamic Augmentation**
>
> - *5. Related to the above, coarse annotations (e.g. Label Styles 1 & 2 in Figure 2) appear to be automatically obtainable from detailed annotations (Label Style 0), using some basic image processing operations (e.g., dilation, then smoothing of the boundary). If so, this may suggest that the coarse annotations actually do not really provide any (significant) new information, and it is possible that with just the detailed annotations for each image, coarse annotations can be generated and used as the additional coarse annotations for the proposed conditional learning. This would however then fall into the (common) use of dynamic augmentation (e.g. minor rotation/scaling, elastic deformation, etc.) preprocessing of the image & annotation labels, in training deep learning classification/segmentation models. Given this, it might be appropriate to explore whether automatically generated coarse segmentations are sufficient to replicate the performance gain currently observed with manually annotated coarse segmentations. If so, this would further inform on the actual source of the performance gain and further reduce human annotation burden.*
>
> Thank you for this very interesting perspective on our work. We conducted additional experiments to evaluate whether similar performance gains can be obtained by dynamic augmentation. We included those experiments in the paper since we believe that they add greatly to the understanding of our method. Our experimental setting is as follows:
>
> 1. We start with the same ISIC training dataset used for the conditioned and baseline models (not the subsets). We substitute coarse annotations with masks that are dynamically augmented from the fine-grained annotations for that image. Next, we draw one annotation randomly when more than one fine-grained annotation is available. We follow your suggestion to first apply dilation and then smoothing the boundary with a Gaussian filter.
> 2. We train the baseline models and the conditioned models on the dataset and evaluate IoU, GED, and area bias.
>
> We find that, while the conditioned models are able to correct for area bias in this setting, segmentation and uncertainty quantification performance are reduced (IoU decreases and GED increases). For the baseline models, area bias and segmentation performance is worse than for all other models. Our takeaways are that the conditioning indeed works for area bias correction, which is confirmed by this experiment.
>
> However, the experiment also supports our reasoning that coarse annotations contain additional information about annotator variability. To clarify this, we have added examples from the ISIC dataset to the paper (see Figure 7 in appendix A.6) that show exactly this: The rough annotations do not always contain the fine-grained annotations. Therefore, it is not possible to capture all the annotator variability when using only fine-grained annotations. Finally, for some training images, fine-grained annotations are unavailable. In that case, a strategy of dynamic augmentation cannot be applied if it is based on fine-grained annotations. These rough annotations can still contain variation that the segmentation uncertainty model can use for learning.
>
> We believe that the suggested experiment adds to the overall understanding of our work and hope that the reviewers find the inclusion into the paper as useful as we do. Of course, the code will be made publicly available.

---

> > ### Comment · Reviewer_f94i · 2022-11-15
> > **Response to Authors**
> >
> > We thank the authors for addressing our comments, particularly on the potential of dynamic augmentation from detailed annotations. While there are of course further methods of generating coarse from detailed annotations, the finding that coarse annotations may encode additional information is enlightening. In view of this and the other responses, our recommendation has been upgraded.

---

> ### Author Response · Authors · 2022-11-11
> **Response to Reviewer f94i - Part 1: Unknown Label Style**
>
> Dear reviewer,
>
> Thank you for your review of our paper and for providing us with valuable questions and feedback. Based on your comments, we conducted additional experiments and have made changes to our paper, which we believe improve the quality. In the following, we address your remarks one by one, and we look forward to further discussing with you.
>
> - *4. A major practical consideration about the proposed method, is the requirement for the label style to be known. For the experiments studied in this work, the datasets have annotations in prior-defined styles (e.g. Figure 2), which are then separately evaluated on as subsets each containing annotations of a single style. It might firstly be informative to evaluate performance where the annotation style is not a priori known.*
>
> This is true. In our motivating examples, however, the label style is known. These correspond to applications in which one either uses two different annotation budgets (expensive = detailed, and cheap = coarse), or in which annotations are created using different tools. We believe that we are tapping into an important and underexplored application even with this requirement already.
>
> However, we acknowledge that our method has additional potential if knowing the label style was not necessary.
>
> Let us first consider the need to know the label styles when training the model. Mathematically, our method transfers directly to a setting where  the label style is a continuous variable. From here on, we could also imagine inferring the label style – either in a supervised or unsupervised manner. However, unsupervised label style inference is nontrivial: the label style can interact with other sources of segmentation variation, and we, therefore, find this to be a (challenging) research problem of its own.
>
> Let us next consider the need for knowing the label style on the test set during evaluation. In Table 3, we investigate how well the models are able to segment in a particular label style by splitting into label style test sets as you describe. In this case, the label style is known a priori and therefore this analysis cannot be used alone for assessing general model performance.
>
> We, therefore, evaluate the models in terms of GED (Table 5) also against a test set of all annotations, and we believe that this is precisely the case you are referring to. Here, it is unclear which label style the conditioned models should use for prediction during test time. Thus, one has to assume a distribution for l in p(y | x,l). In our experiments, we set this to a uniform distribution, which corresponds to no prior knowledge of the label styles in the dataset. We describe this in detail in Appendix A.2.
>
> We apologize that this might not have been clear from the paper and hope that you find our comment useful. Please do not hesitate to comment if this does not address your concerns.

---

### Official Review · Reviewer_9x5M · 2022-10-24

**Confidence:** 4
**Correctness:** 4
**Technical Novelty And Significance:** 4
**Empirical Novelty And Significance:** 4
**Recommendation:** 8

**Clarity, Quality, Novelty And Reproducibility:**

The paper is of great quality, novelty, and reproducibility. The clarity is good except in  Sec.3.1 and Sec.3.2 due to a lack of description of the baseline methods.

**Strength And Weaknesses:**

Strength:
1. The paper presents a clear style of writing.
2. The paper includes detailed experiments and the results are convincing enough to support the hypothesis.
3. The proposed method is simple but surprisingly efficacious and of great potential in increasing the applicability of segmentation uncertainty models.
Weakness:
1. The paper should present a more detailed description of the baseline methods in Sec.3.1 and Sec.3.2.
2. Reasons or comparative experiments suggesting why choosing the modification methods might also be included.
3. The sentence in Sec.1.1 ' We the proposed strategy’ should be ' We proposed strategy' instead.


**Summary Of The Paper:**

The paper suggests that datasets could be heterogeneous since the label generation process might vary systematically, which the paper calls 'label style'. Based on this factor, it further proposes to condition probabilistic segmentation models on label style and to train the models on datasets containing differing label styles. The paper then describes how it modifies Probabilistic U-net and Stochastic Segmentation Networks (SSN) to condition them on label style, and the results of experiments show that compared with Probabilistic U-net and SSN, c-prob. U-net and c-SSNs are able to utilize all data regardless of label style in a meaningful way while introducing less segmentation bias.

**Summary Of The Review:**

The paper proposes a novel hypothesis about the annotation style, and gives a clear description of the hypothesis and method, except for the description of the baseline methods. The experiments are very detailed and the results are convincing.

---

> ### Author Response · Authors · 2022-11-11
> **Response to Reviewer 9x5M**
>
> Dear reviewer,
>
> Thank you for taking the time to review our paper and the assessment of our work as novel and of high quality. We would like to comment on your remarks below.
>
> - *The paper should present a more detailed description of the baseline methods in Sec.3.1 and Sec.3.2.*
>
> To increase the clarity of our description of the baseline methods, we added more detail to sections 3.1 and 3.2 and restructured them. We also added a schematic overview of the baseline models as well as the modifications we make to condition on label style to the appendix, and reference to it at the end of section 3.2. Furthermore, we added formulations of the respective models’ loss functions to the appendix and reference from sections 3.1 and 3.2. We hope that the changes we made in the main paper, as well as the additional sections in the appendix, will enable future readers to better understand all models used; thank you for this valuable comment.
>
> - *Reasons or comparative experiments suggesting why choosing the modification methods might also be included.*
>
> We choose to condition on label style because it gives a natural and practical probabilistic model for the generation of annotations. Having made this modeling choice, we use a standard conditioning method, namely, tiling and concatenating the conditional variable with a feature map, for the two considered models [1,2,3]. Other conditioning methods could be used, e.g., fusing the conditional variable directly with the feature map, as done in [4]. It would be interesting to explore the differences in outcomes between different conditioning methods – thank you for this valuable suggestion – in future work.
>
> - *The sentence in Sec.1.1 ' We the proposed strategy’ should be ' We proposed strategy' instead.*
>
> Please find the updated paper attached, where we corrected this mistake. Thank you for carefully reading our article!
>
> **References:**
>
> [1] Mirza, Mehdi, and Simon Osindero. "Conditional generative adversarial nets." *arXiv preprint arXiv:* 1411.1784 (2014).
>
> [2] Kohl, Simon, et al. "A probabilistic u-net for segmentation of ambiguous images." *Advances in neural information processing systems 31* (2018)
>
> [3] Sohn, Kihyuk, Honglak Lee, and Xinchen Yan. "Learning structured output representation using deep conditional generative models." *Advances in neural information processing systems 28* (2015).
>
> [4] Kassapis, Elias, et al. "Calibrated Adversarial Refinement for Stochastic Semantic Segmentation." *Proceedings of the IEEE/CVF International Conference on Computer Vision.* 2021.

---

> > ### Comment · Reviewer_9x5M · 2022-11-28
> > **Updated response**
> >
> > The author has resolved most of my concerns. I believe this paper is worth being accepted. I will keep my score. Thanks.

---

### Official Review · Reviewer_Pt9b · 2022-10-25

**Confidence:** 3
**Correctness:** 3
**Technical Novelty And Significance:** 3
**Empirical Novelty And Significance:** 3
**Recommendation:** 6

**Clarity, Quality, Novelty And Reproducibility:**

This paper is well-written and easy to follow. The idea is interesting. Experiments are presented with details.

**Strength And Weaknesses:**

Strength
+ An interesting solution to reducing bias caused by aleatoric uncertainty and different label styles
+ Experiments show that the conditioned models can outperform standard models

Weaknesses
- number of image-annotation pairs seems not large
- only discrete label style is included, not the continuous


**Summary Of The Paper:**

This paper proposes a new, updated segmentation uncertainty modeling objective based on the label style and ways to revise the segmentation uncertainty model architecture by including a discrete label style.

**Summary Of The Review:**

The paper presents an interesting idea and good experimental results. However, the evaluation needs improvement.

---

> ### Author Response · Authors · 2022-11-11
> **Response to Reviewer Pt9b**
>
> Dear reviewer,
>
> Thank you for your time reviewing our paper and for providing us with valuable feedback. We would like to address your points one by one.
>
> - *1 number of image-annotation pairs seems not large*
>
> We fully agree that a more comprehensive evaluation of our suggested method, including larger datasets, would strengthen the results. Unfortunately, there are only few publicly available datasets that easily allow studying the effect of different label styles; indeed, even having multiple annotations per image is still rare. However, we hope to help alleviate this problem by making our own ‘style’ version of the PhC dataset publicly available so that future researchers and the community can access other datasets, allowing for more extensive experiments.
>
> When designing this dataset, we prioritized more annotations per image in different styles over having more images – here, the segmentation itself does not need that many data points, which means we can focus on the label style and aleatoric uncertainty modeling.
>
> - *2 only discrete label style is included, not the continuous*
>
> Our work was motivated by real-world examples where annotations naturally vary in a systematic way: on ISIC, labels are generated in three different ways, which are out of our control. On PhC, we have specifically used two different annotation budgets: Expensive (detailed) and cheap (coarse). This does not mean that the continuous case is not interesting, however. Our model carries over directly to the continuous setting, and one could even imagine inferring a continuous family of label styles from the data. However, we find this to be out of the scope of this paper.
>
> - *3 The evaluation needs improvement*
>
> To meet this request along with those of reviewer f94i, we have added an additional experiment (appendix A.6) in which we investigate whether dynamic augmentation of fine-grained annotation can substitute the use of coarse annotations. We believe those results add further to the understanding of our method, and we look forward to further discussing with you.

---

### Decision · Program_Chairs · 2023-01-20

**Decision:**

Accept: poster

**Justification For Why Not Higher Score:**

The empirical evaluations were found to be limited in scope, and concerns were raised about the assumption that label style be known at training time.

**Justification For Why Not Lower Score:**

Empirical evaluations of the method, while limited, still convincingly demonstrate improvements over SOTA.

**Metareview: Summary, Strengths And Weaknesses:**

Hypothesizing that systematic differences in "labeling style" between different manual annotators of image segmentations might lead to bias in models trained on such datasets, the authors propose a method for adjusting the objective function used to train segmentation models, and adapt architectures to condition predictions on such label style differences. The method is shown to improve segmentation accuracy and reduces style bias.

**Note From Pc:**

if the above contains the word "oral" or "spotlight" please see: "oral" presentation means -> notable-top-5% and "spotlight" means -> notable-top-25%. As stated in our emails, we are disassociating presentation type from AC recommendations